OBSERVATION

# L-Form Switching in *Escherichia coli* as a Common $\beta$-Lactam Resistance Mechanism

Aleksandra Petrovic Fabijan,[a,b] David Martinez-Martin,[c,d] Carola Venturini,[a,b] Katarzyna Mickiewicz,[e] Neftali Flores-Rodriguez,[f] Jeff Errington,[e] Jonathan Iredell[a,b,g]

aCentre for Infectious Diseases and Microbiology, Westmead Institute for Medical Research, Sydney, New South Wales, Australia
bSydney Medical School, The University of Sydney, Sydney, New South Wales, Australia
cSchool of Biomedical Engineering, The University of Sydney, Sydney, New South Wales, Australia
dThe University of Sydney Nano Institute, The University of Sydney, Sydney, New South Wales, Australia
eCentre for Bacterial Cell Biology, Bioscience Institute, Newcastle University, Newcastle upon Tyne, United Kingdom
fAustralian Centre for Microscopy and Microanalysis, University of Sydney, Sydney, New South Wales, Australia
gWestmead Hospital, Western Sydney Local Health District, Sydney, New South Wales, Australia

**ABSTRACT** Cell wall deficient bacterial L-forms are induced by exposure to cell wall-targeting antibiotics and immune effectors such as lysozyme. L-forms of different bacteria (including *Escherichia coli*) have been reported in human infections, but whether this is a normal adaptive strategy or simply an artifact of antibiotic treatment in certain bacterial species remains unclear. Here we show that members of a representative, diverse set of pathogenic *E. coli* readily proliferate as L-forms in supratherapeutic concentrations of the broad-spectrum antibiotic meropenem. We report that they are completely resistant to antibiotics targeting any penicillin-binding proteins in this state, including PBP1A/1B, PBP2, PBP3, PBP4, and PBP5/6. Importantly, we observed that reversion to the cell-walled state occurs efficiently, less than 20 h after antibiotic cessation, with few or no changes in DNA sequence. We defined for the first time a logarithmic L-form growth phase with a doubling time of 80 to 190 min, followed by a stationary phase in late cultures. We further demonstrated that L-forms are metabolically active and remain normally susceptible to antibiotics that affect DNA torsion and ribosomal function. Our findings provide insights into the biology of L-forms and help us understand the risk of $\beta$-lactam failure in persistent infections in which L-forms may be common.

**IMPORTANCE** Bacterial L-forms require specialized culture techniques and are neither widely reported nor well understood in human infections. To date, most of the studies have been conducted on Gram-positive and stable L-form bacteria, which usually require mutagenesis or long-term passages for their generation. Here, using an adapted osmoprotective growth media, we provide evidence that pathogenic *E. coli* can efficiently switch to L-forms and back to a cell-walled state, proliferating aerobically in supratherapeutic concentrations of antibiotics targeting cell walls with few or no changes in their DNA sequences. Our work demonstrates that L-form switching is an effective adaptive strategy in stressful environments and can be expected to limit the efficacy of $\beta$-lactam for many important infections.

**KEYWORDS** L-forms, $\beta$-lactams, refractory infections, *Escherichia coli*, antibiotic resistance

Address correspondence to Jonathan Iredell, jonathan.iredell@sydney.edu.au, or Aleksandra Petrovic Fabijan, aleksandra.petrovicfabijan@sydney.edu.au.

The authors declare no conflict of interest.

Bacteria grow at exponential rates under optimal growth conditions when tested in research or diagnostic laboratories (1, 2). However, optimal conditions are rare in nature, where bacteria survive by adjusting their physiology and reducing their growth rates when stressed or starved (3 to 6).

In infection, bacterial growth rates vary significantly and depend primarily on nutrient availability and the host immune response (7). Antibiotics may kill these bacteria rapidly but

sometimes fail to eradicate a small subpopulation that can cause chronic or relapsing infections (8, 9). These cells may survive through genetic adaption to grow normally in the presence of the antibiotics (10) or adapt to tolerate the antibiotic stress and begin normal growth again once conditions improve (11).

Bacterial persisters are specific subpopulations with enhanced tolerance to antibiotics (12). They are growth-arrested bacteria with reduced metabolism that can restart normal growth after stress and have been implicated in antibiotic treatment failure and infection recurrence (12 to 14).

First reported in the 1930s, researchers have recently revisited cell wall-deficient or L-form bacteria with modern molecular biology tools (15 to 21). Metabolically active L-forms have been recently described in macrophages and the urine of patients with recurrent urinary tract infections (UTIs) (20, 21). L-forms may be induced by exposure to cell wall-targeting antibiotics under osmoprotective conditions (17), although some L-forms may tolerate relatively low-osmolality environments (20). Kawai et al. (15) showed that immune effectors such as lysozyme could rescue Gram-positive bacterial viability and protect it from $\beta$-lactam attack by switching into the L-form state (15). L-forms are metabolically active but divide more slowly than exponential-phase cell walled bacteria (CWB) in a manner that is completely independent of the FtsZ-based cell division machinery thought to be essential for normal fission in CWB (16, 17, 22). Without cell walls, L-forms are completely resistant to antibiotics targeting the cell wall (e.g., $\beta$-lactams) (5, 8). Unlike nondividing persister cells, L-forms thrive in the presence of powerful cell walls targeting antibiotics (17, 21), but their cell cycle and growth dynamics have not been well defined.

To date, most studies have been conducted on Gram-positive and stable L-form bacteria, which usually require mutagenesis or long-term serial passages for generation (23). In this study, we show that the L-form is a normal reversible growth state in the archetypal pathogen *E. coli* and define its lag, logarithmic, and stationary phases.

**L-form switching is a common physiological response to cell wall targeting antibiotics in clinical *E. coli* isolates.** It is unclear whether L-forms result from a single process or the final endpoint of a diverse set of processes (17). To characterize the L-form physiology, we developed a double-layer osmoprotective semisolid agar medium to support the aerobic growth of L-forms by modifying the existing L-form medium (LFA) (24). We used this to assess L-form growth (Fig. 1A and B) in 45 genetically distinct *E. coli* isolates from 19 clinically important sequence types (Table S1, posted at https://figshare.com/s/09b4bbc18c62c1d6aadd) in the presence of high concentrations of the $\beta$-lactam-like carbapenem antibiotic meropenem (100 mg/liter). This corresponds to a 100-fold increase of the usual minimal inhibitory concentration (MIC) for this organism and is well above the tested MIC for all of these isolates. L-forms developed quickly and proliferated aerobically in most tested strains (~80%). The majority of L-form cultures (~90%) also quickly reverted to normal rod-shaped CWB within 20 h of meropenem withdrawal (Table S1).

Time-lapse differential interference contrast (DIC) microscopy revealed that bacterial cells quickly lost their regular shape in the presence of high-dose meropenem (MEM) in LFA and increased their surface area about 4-fold (3.95 $\pm$ 1.17), dividing as L-forms within ~5 h (Fig. 1B and C). This asymmetrical scission process yields a morphologically heterogeneous population of *E. coli* cells (Movie S1, posted at https://figshare.com/s/09b4bbc18c62c1d6aadd).

These meropenem-induced L-forms were resistant to growth inhibition by $\beta$-lactam and $\beta$-lactam-like cell wall-active antibiotics such as ceftriaxone (targeting D, d-transpeptidase [DDT] activity of penicillin-binding protein [PBP] 1A/1B), meropenem (PBP2 and L, d-transpeptidase [LDT]), aztreonam (PBP3), amoxicillin (PBP4), and cefoxitin (PBP5/6) (25, 26) (Fig. 1D, right). A range of $\beta$-lactam antibiotics, lysozyme, and macrophages have been shown to induce L-forms (20, 21), but we found that induction of *E. coli* L-forms was more efficient in the presence of meropenem than ceftriaxone, cefepime, or ampicillin. Remarkably, several studies have shown that the essential requirement for PBPs can be fully bypassed by LDTs, replacing the canonical 4–3 cross-links with 3–3 cross-links and leading to broad-spectrum $\beta$-lactam resistance (27, 28). These unusual and often overlooked 3–3 cross-links are present

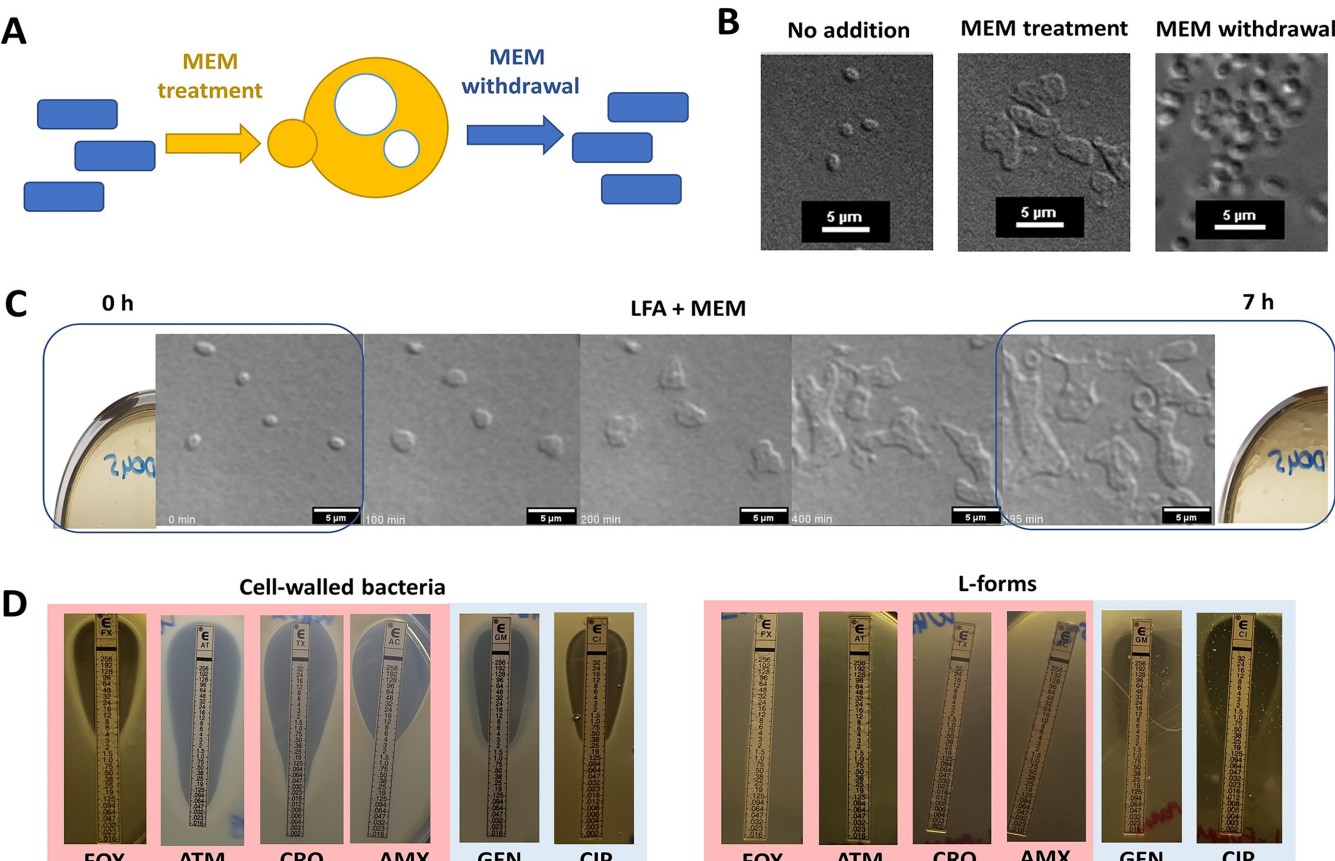

**FIG 1** Meropenem promotes L-form growth from the walled state under aerobic conditions. (A) Model illustrating how pathogenic *E. coli* can switch in and out of the L-form state in response to the antibiotic challenge. (B) *E. coli* L-form strain WH62 switch in the presence of meropenem and reversion to the cell walled state after meropenem withdrawal. (C) Time-lapse DIC microscopy of WH62 L-forms; individual micrograph frames are extracted from Movie S1, posted at https://figshare .com/s/09b4bbc18c62c1d6aadd. (D) Susceptibility of *E. coli* L-forms to β-lactams, aminoglycosides, and fluoroquinolones by standard Etest. MEM, meropenem; FOX, cefoxitin; ATM, aztreonam; CRO, ceftriaxone; AMX, amoxicillin; GEN, gentamicin; CIP, ciprofloxacin.

in a smaller but significant proportion of the bacterial cell wall (e.g., 3 to 15% in *E. coli* cells, mostly depending on their genetic makeup and growth phase) (29, 30), and might explain incomplete induction of the L-forms in the presence of ampicillin and cephalosporins observed in this study. On the other hand, efficient L-form switching was evident in the presence of meropenem and might be attributable to its dual action and rapid inactivation of both D,d-transpeptidase (PB2) and L,d-transpeptidase in *E. coli* (31, 32).

Carbapenem susceptibility of CWB was identical before and after L-form transition (Table S2) despite the absolute carbapenem and β-lactam resistance of their L-form state. The widely used fluoroquinolone antibiotic ciprofloxacin, targeting DNA gyrase and topoisomerase enzymes, remained a potent inhibitor of L-form growth with an MIC of <0.25 mg/L (33) in a modified Etest; aminoglycoside susceptibility was also retained in L-forms developed from gentamicin (aminoglycoside)-susceptible CWB (MIC <2 mg/L) (Fig. 1D, left).

To characterize growth rate and metabolic activity, we used semisolid agar in a 24-well plate supplemented with MEM (100 mg/L) and 2,3,5-triphenyltetrazolium chloride (TTC) as a redox indicator, measuring optical density at 540 nm ($OD_{540}$) of *E. coli* J53 (a well-characterized *E. coli* K-12 derivative) (34) and WH62 (clinical isolate) (35).

Both grew more slowly as L-forms than as CWB, with an initial lag phase (Fig. 2A) of ~80 and ~190 min for WH62 and J53, respectively, as L-forms developed from CWB in osmoprotective LFA with MEM. Growth rates increased after this lag phase, while CWB controls were completely inhibited in nonosmoprotective media (Fig. 2A, left). L-form population growth then appeared to enter a stationary phase after ~500 min of incubation, with evidently reduced metabolic activity (Movie S1, posted at https://figshare

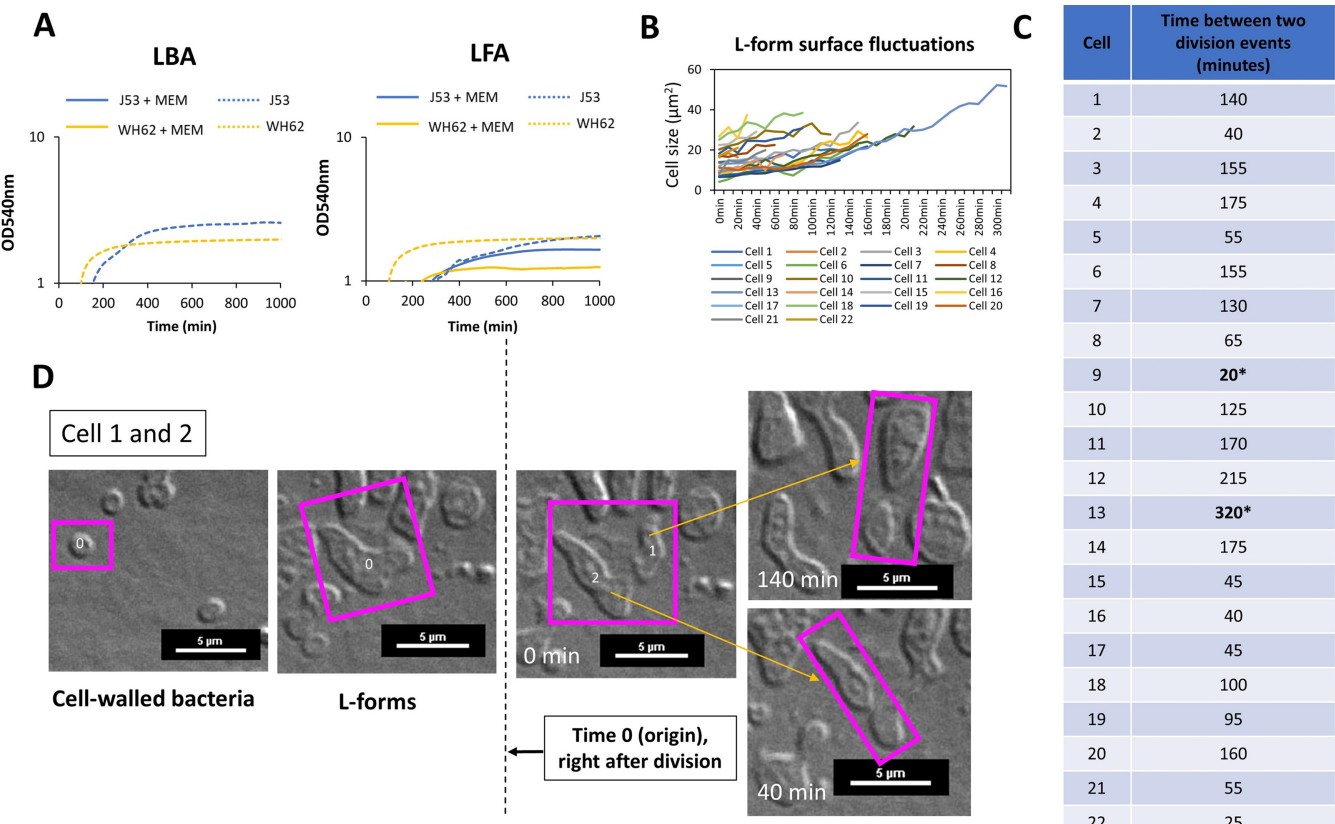

**FIG 2** WH62 *E. coli* L-form growth kinetics and proliferation rates in LFA medium supplemented with the antibiotic meropenem. (A) Growth curves of J53 (ST10) and WH62 (ST127) L-forms. (B) Bacterial cell area fluctuations between division measured in 22 cells. (C) Length of L-form cell cycle measured in 22 different L-form cells. (D) Mode of cell division of L-forms, division by budding; see also Movie S1, posted at https://figshare.com/s/09b4bbc18c62c1d6aadd. Time 0 (origin) indicates the first division event in the L-form state.

.com/s/09b4bbc18c62c1d6aadd, and Fig. 2A). Time-lapse microscopy revealed heterogeneous growth during the L-form exponential phase in which shape deformation and an increase of cell surface area preceded asymmetrical scission and the emergence of new progeny (Fig. 2B). Imaging revealed a wide variation in the periodicity of CWB to L-form switching around an average of 114 ($\pm$75 min) in 22 independent cells (e.g., Fig. 2C and D; Data set S1, posted at https://figshare.com/s/09b4bbc18c62c1d6aadd), generally consistent with the observed lag phase determined for L-form populations overall (Fig. 2A). Intracellular vesicles were evident after prolonged incubation in aerobic conditions (>16h) (Fig. S2).

The genomes of four genetically distinct (Fig. S1 and Table S3, posted at https://figshare.com/s/09b4bbc18c62c1d6aadd) CWB revertants (i.e., after transition to L-form and back) differed from parent CWB strains (i.e., before the transition to L-form) in only one of the four pairs tested (B36_rev compared to *E. coli* B36), in which single nucleotide variants arose mainly in loci encoding surface-presented molecules, including common bacteriophage receptors (e.g., capsule) (Table S4, posted at https://figshare.com/s/09b4bbc18c62c1d6aadd; accession number: PRJNA764821).

Predating viruses (bacteriophages or phages) are a common threat to bacterial populations and are now increasingly used for therapy, including of *E. coli* infection and often in combination with $\beta$-lactam antibiotics (2, 36). We therefore investigated predation of established L-forms by representatives of the ubiquitous and therapeutically important T4-like myoviruses (vB_EcoM_OMNI2, Eco2; and vB_EcoM_OMNI12, Eco 12) and of the less common V5-like phages (vB_EcoM_OMNI6, Eco6) (Fig. 3A). While Eco6 lysed L-forms and CWB equally (Fig. 3B, top), T4-like Eco2 and Eco12 failed to propagate in L-forms of strains in which they are obligately lytic of CWB

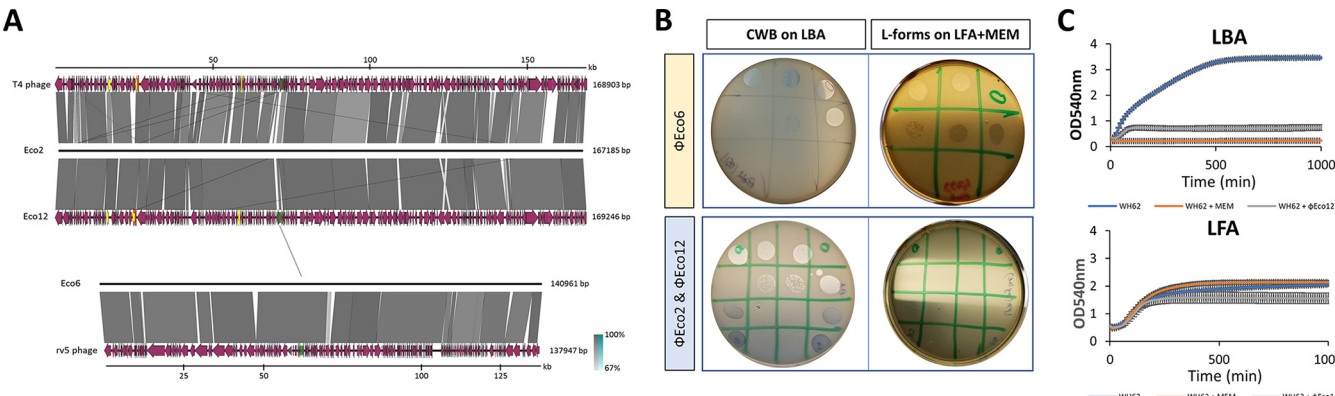

**FIG 3** Effects of the L-form switch on susceptibility to myoviruses. (A) Comparative analysis of phage genomes. Schematics show the genomic organization of phages vB_EcoM_OMNI-2 (Eco2), 6 (Eco6), and 12 (Eco12; GenBank OL362041) compared to available reference genomes (T4-like phage NC_000866.4 and V5-like phage DQ832317.1) (left). (B) Phage susceptibility of *E. coli* L-forms using standard spot assay and modified LFA; WH62 and JIE4799 meropenem-induced L-forms displaying resistance (no lysis) to T4-like phages (Eco2 and Eco12) (bottom right) and sensitivity to V5-like phages (Eco6) (top right), respectively. Control involved cell-walled counterpart on standard LBA without meropenem (left top and bottom) lysed by all three phages. (C) Growth curves of walled bacteria (WH62 on LBA) and L-forms (L-WH62 on LFA supplemented with meropenem) in the presence of Eco12 phage at MOI 1.

forms (Fig. 3B, bottom), and L-form growth was unaffected (Fig. 3C). We could not induce L-form switching by exposing CWB to T4-like phages in LFA (Eco12, on three separate occasions), although this has been described for other bacteria (37, 38).

$\beta$-lactam antibiotics remain one of the most commonly prescribed drug classes (39, 40) but often fail in severe and refractory infections despite demonstrated efficacy *in vitro* (41, 42). L-forms may be an important contributor to bacterial virulence and to the failure of antibiotic treatment with $\beta$-lactams and related antibiotic classes.

Our data indicate that L-forms are an effective and probably ancient stress response that appears to be the norm in *E. coli* populations, exhibiting nonbinary cell growth with well-defined lag, log, and stationary phases.

**Bacterial strains.** The potential for L-from growth was tested in a wide range of *E. coli* strains (*n* = 45), including a multidrug-resistant dominant clone sequence type (ST) 131 (clade A, B, and C). The testing also involved strains belonging to other clinically important STs (*n* = 19), which are listed in Table S1. A total of 15 genetically distinct strains were selected for further testing, which included testing of meropenem susceptibility in revertants and phage susceptibility (Table S2, posted at https://figshare.com/s/09b4bbc18c62c1d6aadd).

**Growth conditions.** *E. coli* isolates were grown on Brilliance GBS Agar/Oxoid (Thermo Fisher Scientific) and in Lysogeny broth (LB-Miller, Becton, Dickinson, France). Bacterial L-forms were induced in osmoprotective LFA as described previously (24). When necessary, antibiotics and supplements were added at the following concentrations: meropenem or MEM (100 $\mu$g/mL) and 2,3,5-triphenyl-2H-tetrazolium chloride or TTC (5%). To assess L-forms switching in 45 genetically distinct *E. coli* clinical strains, we used a modified double-layer osmoprotective LFA (24). Briefly, 300$\mu$L of a bacterial suspension was grown to exponential phase at 37°C, with shaking at 225 rpm (revolutions per minute) and MEM was added to 4 mL top agar, gently homogenized, and poured into 90-mm petri dish previously prepared with 10 mL bottom agar. The plates were gently swirled, dried for 10 min at room temperature, and then inverted and incubated at 37°C overnight. The control included (i) growth of typical cell-walled bacteria (CWB) on standard hypotonic Luria-Bertani agar (positive control; LBA); (ii) growth inhibition of cell-walled forms by high dose of cell wall-targeting antibiotics, in standard hypotonic LBA (negative control; LBA+MEM) (Fig. 4). Reversion to the cell wall state was demonstrated by plating out L-forms on both LFA and LBA (<10% survived) without antibiotics.

**Antibiotic susceptibility assays.** MICs of meropenem in revertants was assessed using microbroth-dilution protocol as previously described and interpreted according to the Clinical and Laboratory Standard Institute (CLSI) (1). The Etest (bioMérieux, USA) was conducted

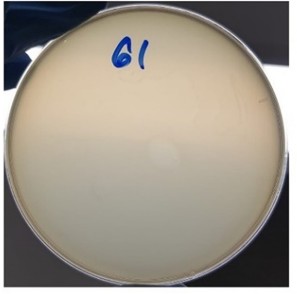
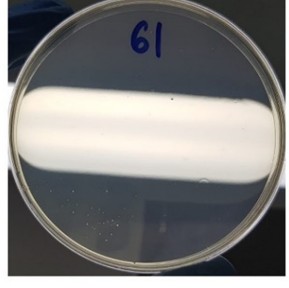
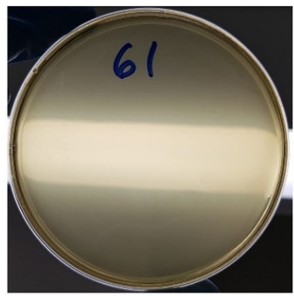

**CWB growth** on LBA
- positive control

**No growth** on LBA+MEM (100µg/ml)
– negative control

**L-form growth** on LFA+MEM
(100µg/ml)

**Fig 4** The double-layer method was used to test the L-form switching in pathogenic strains of *E. coli*.

according to the manufacturer's instructions on Mueller-Hinton agar (MHA) for walled bacteria and on LFA for L-forms with certain modifications. Bacterial lawns of WH62 and J53 were prepared using the double-layer method described above. After applying the Etest strip, the plates were incubated for 24 h aerobically. The MIC of the antibiotic was read directly from the scale printed on the Etest strip at the point of intersection between the bacterial growth zone and the strip. Susceptibility to antibiotics (amoxicillin, ceftriaxone, cefepime, and cefoxitin) was determined according to internationally accepted CLSI breakpoints (1). An *E. coli* ATCC25922 was used as a control.

**Microscopic imaging and growth kinetics.** Sample preparation for time-lapse differential interference contrast (DIC) microscopy has been done as previously described with a slight modification (43). Instead of a standard imaging medium, LFA supplemented with meropenem was used. DIC microscopy images were acquired at 37°C on a Nikon Eclipse Ti-E motorized inverted microscope with Perfect Focus using a Nikon 100 × 1.45 pathogenic isolates of *E. coli* PlanApo Lambda Objective (Nikon Instruments). Images were captured using a Nikon DS-Qi2 monochrome camera at 5-min intervals for up to 16 h using the NIS-Elements software (Laboratory Imaging s.r.o.). DIC illumination was achieved using Nomarski prisms. Pictures and videos were prepared for publication using Huygens Professional version 19.04 (Scientific Volume Imaging, The Netherlands, http://svi.nl) and ImageJ (http://rsb.info.nih.gov/ij) (44).

**Isolation of *Escherichia coli* specific phages.** Bacteriophages vB_EcoM_OMNI2, vB_EcoM_OMNI6, and vB_EcoM_OMNI12 targeting pathogenic isolates of *E. coli* were isolated from sewage and pond water samples respectively collected in the Greater Sydney District (Sydney, NSW, Australia) during 2019. Specimens were clarified by filtration through 0.45-$\mu$m and 0.22-$\mu$m filters. Isolation of bacteriophages was performed using an enrichment procedure (45) where single plaques were picked and purified as previously described (46). High-titer stocks were prepared by propagating bacteriophages over several double-layer plates washed in SM buffer (50 mM Tris-HCl, 8 mM MgSO4, 100 mM NaCl, pH 7.4), filtered through a 0.22-$\mu$m filter and precipitated with NaCl and PEG8000 (46). The concentration as plaque-forming units per mL (PFU/mL) was determined by spotting 10 $\mu$L of 10-fold serial dilutions onto a double layer of the target bacteria (46). High-titer ($\geq$1,010 PFU/mL) bacteriophage stocks were stored at 4°C.

**Phage susceptibility.** Phage-susceptibility testing was performed using a traditional plaque or a double-layer agar method as previously described (46). When testing phage susceptibility in L-forms, instead of standard LBA medium, LFA supplemented with meropenem was used. Inhibition of cell-walled and L-form bacterial growth was determined as described previously (24), with a modification that included LFA supplemented with meropenem to support L-form growth.

**Data availability.** All data generated or analyzed during this study are included in this article and its supplemental material (posted at https://figshare.com/s/09b4bbc18c62c1d6aadd).

Whole-genome sequencing data are available on NCBI under the BioProject accession number PRJNA764821 and GenBank number OL362041.

## ACKNOWLEDGMENTS

We are grateful to Sally Partridge for advice on experimental design. We thank We Nouri Ben Zakour for her help with phage genome sequencing data and staff at the Pathogen Genomics Unit, Westmead Hospital for their technical advice and sequencing support. The authors acknowledge the technical and scientific assistance of Sydney Microscopy and Microanalysis, the University of Sydney node of Microscopy Australia.

A.P.F. is supported by the Office for Health and Medical Research (New South Wales, Australia) Phage Therapy Fellowship. This work was funded by a National Health Medical Research Council (Australian Government) Investigator Grant (Iredell_APP1197534).

A.P.F. and J.I. conceived the study and designed the main experimental plan. A.P.F., J.I., J.E., and K.M. analyzed the data. A.P.F. and J.I. wrote the paper. A.P.F. developed a double-layer plaque and microtiter assay that supports the L-forms growth *in vitro*. A.P.F., D.M.-M., and N.F.-R. designed the L-form microscopy experiments. N.F.-R. and A.P.F. performed imaging experiments and data image processing. C.V. analyzed whole-genome sequencing data of *E. coli* and revertants. A.P.F. performed all experiments. All authors were involved in reviewing and editing the final manuscript.

We declare no competing interests.

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
