## [Reviewer comments · Microbiology Spectrum]

Microbiology Spectrum

L-form switching in *Escherichia coli* as a common beta-lactam resistance mechanism

Aleksandra Petrovic Fabijan, David Martinez-Martin, Carola Venturini, Katarzyna Mickiewicz⁵, Neftali Flores-Rodriguez, Jeffery Errington, and Jonathan Iredell

Corresponding Author(s): Aleksandra Petrovic Fabijan, Westmead Institute for Medical Research

Review Timeline:

Submission Date:	August 3, 2022
Editorial Decision:	August 9, 2022
Revision Received:	August 15, 2022
Accepted:	August 19, 2022

Editor: Gyanu Lamichhane

Reviewer(s): The reviewers have opted to remain anonymous.

Transaction Report:

DOI: <https://doi.org/10.1128/spectrum.02419-22>

August 9, 2022

Dr. Aleksandra Petrovic Fabijan
Westmead Institute for Medical Research
176 Hawkesbury Road
Westmead, New South Wales 2145
Australia

Re: Spectrum02419-22 (L-form switching in Escherichia coli as a common beta-lactam resistance mechanism)

Dear Dr. Aleksandra Petrovic Fabijan:

Thank you for submitting your manuscript to Microbiology Spectrum. As you will see your paper is very close to acceptance. Please modify the manuscript along the lines I have recommended. As these revisions are quite minor, I expect that you should be able to turn in the revised paper in less than 30 days, if not sooner. If your manuscript was reviewed, you will find the reviewers' comments below.

Based on prior review and my own review of the revised manuscript, discussion of the prior literature relevant to mechanisms of resistance to b-lactam exposure is inadequate. As the prior reviewer noted, LD-transpeptidases also play a role in resistance to certain b-lactams. Slow growing E. coli, changes to cell wall physiology including LD-transpeptidases mediated 3-3 linkages have been previously described (eg, Tuomanen E, Cozens R. Changes in peptidoglycan composition and penicillin-binding proteins in slowly growing Escherichia coli. J Bacteriol. 1987 Nov;169(11):5308-10. doi: 10.1128/jb.169.11.5308-5310.1987. PMID: 3312172; PMCID: PMC213942.) Including, this and more recent papers describing potential roles of LD-transpeptidases in the discussion will make the manuscript more comprehensive.

When submitting the revised version of your paper, please provide (1) point-by-point responses to the issues I raised in your cover letter, and (2) a PDF file that indicates the changes from the original submission (by highlighting or underlining the changes) as file type "Marked Up Manuscript - For Review Only". Please use this link to submit your revised manuscript. Detailed instructions on submitting your revised paper are below.

Link Not Available

Sincerely,

Gyanu Lamichhane

Reviewer comments:

Preparing Revision Guidelines

- point-by-point responses to the issues I raised in your cover letter
- Upload a compare copy of the manuscript (without figures) as a "Marked-Up Manuscript" file.
- Each figure must be uploaded as a separate file, and any multipanel figures must be assembled into one file.
- Manuscript: A .DOC version of the revised manuscript

- Figures: Editable, high-resolution, individual figure files are required at revision, TIFF or EPS files are preferred

Please return the manuscript within 60 days; if you cannot complete the modification within this time period, please contact me. If you do not wish to modify the manuscript and prefer to submit it to another journal, please notify me of your decision immediately so that the manuscript may be formally withdrawn from consideration by Microbiology Spectrum.

August 19, 2022

Dr. Aleksandra Petrovic Fabijan
Westmead Institute for Medical Research
176 Hawkesbury Road
Westmead, New South Wales 2145
Australia

Re: Spectrum02419-22R1 (L-form switching in Escherichia coli as a common beta-lactam resistance mechanism)

Dear Dr. Aleksandra Petrovic Fabijan:

Your manuscript has been accepted, and I am forwarding it to the ASM Journals Department for publication. You will be notified when your proofs are ready to be viewed.

Sincerely,

Gyanu Lamichhane
Editor, Microbiology Spectrum
